# Advancements in Focal Amplification Detection in Tumor/Liquid Biopsies and Emerging Clinical Applications

**DOI:** 10.3390/genes14061304

**Published:** 2023-06-20

**Authors:** Aram Arshadi, Doron Tolomeo, Santina Venuto, Clelia Tiziana Storlazzi

**Affiliations:** Department of Biosciences, Biotechnology and Environment, University of Bari Aldo Moro, 70125 Bari, Italy; aram.arshadi@uniba.it (A.A.); doron.tolomeo@uniba.it (D.T.); santina.venuto@uniba.it (S.V.)

**Keywords:** focal amplification, oncogene, double minute, homogeneously staining region, episome

## Abstract

Focal amplifications (FAs) are crucial in cancer research due to their significant diagnostic, prognostic, and therapeutic implications. FAs manifest in various forms, such as episomes, double minute chromosomes, and homogeneously staining regions, arising through different mechanisms and mainly contributing to cancer cell heterogeneity, the leading cause of drug resistance in therapy. Numerous wet-lab, mainly FISH, PCR-based assays, next-generation sequencing, and bioinformatics approaches have been set up to detect FAs, unravel the internal structure of amplicons, assess their chromatin compaction status, and investigate the transcriptional landscape associated with their occurrence in cancer cells. Most of them are tailored for tumor samples, even at the single-cell level. Conversely, very limited approaches have been set up to detect FAs in liquid biopsies. This evidence suggests the need to improve these non-invasive investigations for early tumor detection, monitoring disease progression, and evaluating treatment response. Despite the potential therapeutic implications of FAs, such as, for example, the use of HER2-specific compounds for patients with *ERBB2* amplification, challenges remain, including developing selective and effective FA-targeting agents and understanding the molecular mechanisms underlying FA maintenance and replication. This review details a state-of-the-art of FA investigation, with a particular focus on liquid biopsies and single-cell approaches in tumor samples, emphasizing their potential to revolutionize the future diagnosis, prognosis, and treatment of cancer patients.

## 1. Introduction

Cancer is a genetic disease caused by multiple events, including the loss of tumor suppressor genes and the activation of proto-oncogenes [1]. Proto-oncogenes are usually activated through several mechanisms, such as point mutations, chromosomal rearrangements (leading to position effects [2] or gene fusions [3]), or high-copy number amplification [4]. Oncogene amplification is critical in tumorigenesis and tumor progression, as it provides cell growth advantage and resistance to drug treatment [5]. In cancer genomes, oncogene amplification mainly occurs as focal amplifications (FAs) of the genomic regions, i.e., amplicons of chromosomal segments smaller than 20 Mb in size [6]. They are present in at least 30 different cancer types, including breast, colon, and lung cancers, and are often associated with a worse patient prognosis [7,8]. Recent advances in molecular investigations have enhanced our understanding of FAs’ properties and functions, emphasizing their relevance as therapeutic targets and introducing new approaches and techniques for recognizing them and analyzing their structure. For example, therapeutic strategies for reducing the FA levels in cancer cells could downregulate the expression of the amplified genes involved in aggressive phenotypes and drug resistance in advanced cancers [9]. Moreover, FAs have recently been studied extensively for their potential as diagnostic and prognostic tools, also considering their occurrence as circulating DNA into the peripheral blood system [10,11].

Here, we review FA architectures and genesis mechanisms. We will focus on methods and tools for detecting genomic amplification in tumor and liquid biopsies, highlighting their potential implications for cancer diagnosis, treatment, and patient follow-up.

## 2. Cytogenetic Manifestation of Genomic Amplifications and Mechanisms of Their Genesis

### 2.1. Types of Genomic Amplification

In cancer genomes, FAs may occur either extra- or intra-chromosomally [12]. The former are mainly present as episomes and double minute chromosomes (DMs), also known as extrachromosomal DNAs (ecDNAs) [9,13,14]; the latter are visible as homogeneously staining regions (HSRs) (Figure 1a) [12].

Episomes can range in size up to 1 Mb, so they cannot be observed using a conventional cytogenetic analysis as ecDNA elements. Despite their small size, they have been shown to play an interesting role in oncogene activation, not only because they can harbor and amplify oncogenes, such as *MYCN* in medulloblastomas [13], but as they can also generate fusion genes via their circularization. For instance, Graux et al. described an amplification of the *NUP214::ABL* fusion on episomes in T-cell Acute Lymphoblastic Leukemia (T-ALL) patients [15].

Episomes may represent the first step of genomic amplification, originating after the excision/deletion of the corresponding genomic segment from the original chromosomal location and its subsequent circularization. This mechanism, known as the “episome model”, was initially proposed in 1988 by Carroll et al., who suggested that episomes can gradually expand until they become DMs, which could alternatively be integrated into chromosomes to generate HSRs [16].

DMs are a type of FA first described over five decades ago [17]. Originally referred to as “minute chromatin bodies”, they include circular DNA elements containing tandemly arrayed genomic segments (amplicons), ranging from 1 to 3 Mb in size [14]. They can be observed through a conventional cytogenetics analysis. 

DM amplicons are described as harboring known oncogenes. They are the primary source of genomic amplification in neoplasia and observed in nearly half of human cancers, representing a driving force toward the accelerated evolution of tumors [18]. The main impact of these ecDNA structures consists of the activation of embedded oncogenes, which increases their expression level for a gene copy number increase and their peculiar chromatin topology. It has been demonstrated that ecDNA nucleosomal organization is less compacted than chromosomal DNA, leading to enhanced DNA accessibility for the transcriptional machinery at oncogene promoters [14]. Furthermore, an increased number of ultra-long range interactions within ecDNA active chromatin has also been observed, indicating effects on the transcription of the genes that are typically distant on chromosomes [14]. Another interesting consequence of amplicon arrangement within ecDNA is the genesis of amplified intra- and inter-chromosomal fusion genes, potentially impacting tumorigenesis and tumor progression [19] due to the remodeling of the transcriptional landscape associated with ecDNAs.

DMs are also involved in determining intratumoral heterogeneity, which is crucial to causing drug resistance and a shorter overall survival in cancer patients [20] for two main reasons. The former is related to differences in the internal amplicon architecture. An integration of cytogenetic and molecular approaches shows the coexistence of different amplicon types, diverging in size and sequence composition, even when they show the same embedded oncogene in the same tumor cell population [21,22]. The latter is due to the ecDNA copy number in cells, which is a consequence of the acentric structure of ecDNAs, leading to the random segregation of DMs in daughter cells following a non-mendelian inheritance mechanism [23]. Interestingly, some authors have described a tethering of ecDNAs at chromosome extremities during anaphase, called “hitchhiking”, which drives their unequal segregation and distribution among the cells originated by cell division [24,25,26] (Figure 1b). Furthermore, if a single DM attaches to the tips of sister chromatids, their segregation at anaphase could form DNA bridges that could end with breakage upon cytokinesis [24,25], leading to differences in the amplicon copy numbers in the dividing cells and contributing to their genetic heterogeneity [18]. 

Despite the established acentric nature of DMs, we documented the emergence of ectopic centromeres (neocentromeres) in them, increasing their size towards small ring chromosomes in a portion of tumor cells in the bone marrow of Acute Myeloid Leukemia (AML) patients with *MYC*-DM amplifications [21]. We speculate that centromere recruitment at ecDNAs could augment the selective advantage of tumor cells carrying genomic amplifications, ensuring ecDNAs’ mitotic stability upon cell division (Figure 1c). Other examples of neocentromere seeding in the literature are mainly in regard to well-differentiated liposarcomas on ring and rod-shaped chromosomes harboring high-level amplifications, hence stabilized by the ectopic centromeres [27,28].

DMs are described to evolve towards HSRs in vivo and in vitro through chromosome reintegration [29,30].

HSRs are chromosomal regions harboring genomic amplifications that display an atypical banding pattern after trypsin-Giemsa staining (Figure 1c). Their amplicons might be head-to-tail oriented when located within different chromosomal sites from that of the target oncogene [16], as they might be originated by episome/ecDNA integration after their genesis and amplification. Alternatively, HSR amplifications located close to the original oncogene site are formed via the breakage-fusion-bridge (BFB) cycle model and have a head-to-head orientation [22] (see Section 2.2). 

Both ecDNAs and HSRs can coexist in various types of tumors, as revealed by re-assessing large-scale DNA sequencing datasets. Glioblastoma (GBM) and sarcoma tumors have the highest frequency of ecDNAs (60% and 47%, respectively) [7]. HSRs are commonly found in specific cancer types, such as lung squamous (32%) and bladder carcinomas (28%) [7,30]. Several oncogenes are amplified on ecDNAs and HSRs, including *MDM2*, *MYC*, *EGFR*, *CDK4*, and *ERBB2* [7,31]. It is worth noting that HSRs are less accessible at the chromatin level than ecDNAs. Therefore, the same high-copy number increase leads to a higher expression of an oncogene if it is amplified on ecDNAs rather than HSRs [14]. 

### 2.2. Mechanisms of FA Emergence

It has been well established that the genomic amplification process starts after double-strand breaks (DSBs), which trigger a copy number increase in one or more oncogenes following different mechanisms. DNA replication stress due to carcinogen and pathogen exposure and a failure in the DNA damage repair pathway could produce DSBs [32]. As summarized in Figure 2, three main mechanisms are involved in the genesis of DMs and HSRs: BFB cycles, the extra replication (or loop-formation)–excision–amplification model (also known as the “episome” model), and chromothripsis.

#### 2.2.1. BFB Cycles Model

In this model, the initial DSB usually locates telomerically to a target oncogene due to the occurrence of downstream common fragile sites [33] or collapsed replication-stalled forks [34]. The DSB is then followed by the fusion of two broken ends of sister chromatids, forming a dicentric anaphase bridge [35]. The products of the asymmetric breakage of the dicentric bridge are broken down in daughter cells, showing an inverted duplication and deletion [36]. If the chromosome with the inverted duplication undergoes subsequent BFB cycles, it will display FAs as inverted duplications [37]. According to the model, once the BFB cycle has been initiated, the consecutive breakages of the anaphase bridges must occur near the first breakage location for HSR formation to occur [38] (Figure 2a). 

Alternatively, dysfunctional telomeres could trigger BFB cycles after their joining via non-homologous end-joining (NHEJ) to create dicentric chromosomes among the sister chromatids [39]. Several gene amplifications or genomic instability events have been attributed to this model, including amplifications of the *DHFR*, *AMPD2*, *CCND1*, *MDM4*, *EGFR*, *AKT3*, and *ERBB2* oncogenes [7].

#### 2.2.2. Episome Model

Episome genesis was first described by Wahl and colleagues in 1987 [40]. Accordingly, replication fork stalling elicits its collapse and the replication bubble subsequently falls off the chromosome, circularizes, and transforms into an autonomously replicating episome [16,41] (Figure 2b). Additionally, the autonomous replication and recombination of episomes lead to the formation of ecDNAs or their integration into different chromosomes to form HSRs [16]. The higher frequency of these elements in cancer cells compared to normal cells shows that they may contribute to cell transformation and tumor progression, increasing genomic instability [42]. For example, our group reported that *MYC*-containing ecDNAs in AML patients could have originated from this mechanism. In Storlazzi et al. (2006), 23 out of 30 AML-reported patients showed deletions of the 8q24 amplified region on one of the two chromosomes 8 and, in one case, the sequencing of the DM and del(8q) junctions revealed their correspondence [43]. Furthermore, in L’Abbate et al. (2018), 11 out of 23 AML patients investigated via an SNP array and whole-genome sequencing (WGS) displayed the heterozygous deletion of the amplified region on DMs, rings, and HSRs [21]. This evidence is in perfect agreement with the episome model [43].

The episome model could also be applied to derivative chromosomes originated via chromosomal translocations. Barr et al. described a “translocation-amplification” mechanism, in which amplification events occur after translocations and are frequently triggered by exogenous stimuli [44]. The DNA repair system can excide fusion sequences that could generate DMs and/or HSRs [45]. In a few studies, through this mechanism, amplifications of the *MYC*, *ATBF1*, *HMGIC*, and *MDM2* oncogenes have been observed [46,47].

#### 2.2.3. Chromothripsis

The word “chromothripsis”, coined by Stephens et al. in 2011, indicates a single-step catastrophic event in which one or a few chromosome/s is/are broken into thousands of tiny fragments that are randomly rejoined, resulting in a profound genomic rearrangement that affects the entire chromosomes or chromosomal arms [48]. Chromothripsis is described to drive ecDNA genesis by originating episomic fragments that undergo extra-chromosomal amplification (Figure 2c). Furthermore, a telomeric fusion of the chromosomes involved in a chromothripsis event has been linked to the genesis of complex ecDNA in some cancers, including medulloblastomas with *TP53* mutations [49] and small-cell lung cancer (SCLC) [50,51]. Ly et al., by selectively inactivating the Y chromosome centromere, also established that chromothripsis induced by mitotic errors could produce ecDNA [52].

## 3. Methods and Bioinformatic Tools to Detect Genomic Amplification in Tumor Samples

### 3.1. Definition of Amplicon Structure at the DNA Level

The detection and analysis of FAs have become increasingly important for understanding tumor biology and devising targeted therapies. Several wet methods and bioinformatic tools have been developed, which are often used as integrated multi-technique approaches to investigate the amplicon unit structure, chromatin architecture, and compaction status related to the embedded oncogene expression. Here, we provide an overview of these methods and tools from the earliest to most recent ones (Figure 3).

#### 3.1.1. Detection of DMs/HSRs

Early studies on FAs relied on electron microscopy (EM), including scanning electron microscopy (SEM) and transmission electron microscopy (TEM), providing the first visual evidence of FAs in cancer cells and being particularly useful for visualizing the circular nature of DMs [57]. Subsequently, optical microscopy techniques, such as brightfield, phase contrast, and fluorescence microscopy, have been employed to observe FAs and their localization within cells [14]. Although these techniques allow for high-resolution single-cell analyses of the DM number, cytogenetic structure, spatial location into the nucleus, and position relative to cell chromosomes, they are not informative about the DM gene content, chromatin status, and gene expression regulation. For these reasons, to the best of our knowledge, their use in FA studies is presently minimal. Following this, fluorescence in situ hybridization (FISH) has been used to identify ecDNAs/HSRs and detect specific gene amplifications, offering valuable insights into the presence and localization of these elements within tumor cells [58]. FISH analyses with probes specifically designed for amplified genomic regions have added important information related to the mechanisms of gene amplification in various cancer types as, in some cases, a “scar” on the chromosome of origin corresponding to the amplified sequences on the ecDNA/HSR has been found [21,43,53,59,60]. Moreover, FISH allows for the mapping of HSR insertion sites in some tumor types, showing no particular sequence tag at their locations, supporting an NHEJ process [22,61]. Finally, due to their application at the single-cell level, FISH assays can also unveil the heterogeneity of the amplified regions among different cells of the same sample [21]. Compared to microscopy-based assays, an important advantage of the FISH technique is the possibility of detecting FAs in interphase cells (i-FISH), which are often more abundant in cancer samples than metaphases. Zakrzewski et al. exploited i-FISH to develop a pipeline for the automated evaluation of the *ERBB2* amplification status in breast and gastric cancer [62]. The analytical resolution of FISH is related to the size of the probes used, ranging between whole chromosomes (for the whole-chromosome paintings) and 40–50 kb of the fosmid probes [63]. Due to the probe size, FISH does not allow for FA characterization at a single nucleotide resolution level; hence, it is not adequate for characterizing the structural variations joining the different genomic regions of the amplicons or for determining the exact fusion junction generating a chimeric gene.

The continuous development of advanced methods, such as ultra-high-resolution microscopy technologies, for example, super-resolution three-dimensional structured illumination microscopy (3D-SIM), have further enabled the visualization of structures and molecules with a high spatial resolution [14].

In addition to wet-lab techniques, image analysis tools have also been developed to analyze the FA occurrence in cells. In 2017, ECdetect, a semi-automated image analysis software package, was designed to detect and quantify FAs by analyzing DAPI-stained metaphase images. Turner et al. utilized this tool to reveal the presence of ecDNAs in nearly half of the 17 cancer types studied, highlighting their role in accelerating cancer progression and increasing intratumoral heterogeneity [18]. However, ECdetect can only report FAs occurring as ecDNAs, as it does not reveal HSRs and does not provide information about their genomic structure. Furthermore, Rajkumar et al. developed ecSeg, a tool that employs the U-Net machine learning algorithm for analyzing DAPI-stained DNA images, enabling the identification of FAs and localization of oncogene amplification at the single-cell level [54]. Integrating FISH-stained metaphase images, with a maximum of two hybridized probes, ecSeg has the advantage of detecting both ecDNAs and HSRs at the single-cell level [64]. A limitation of both ECdetect and ecSeg lies in the need for cells in metaphase; for example, in their work, Turner et al. analyzed at least 20 metaphases for each sample under study, a number not always easy to reach [18].

#### 3.1.2. Amplicon Architecture Definition

For the reconstruction of the internal structure of FAs, Southern Blotting and FISH assays [43,64] have been integrated with long-range PCR approaches to map the amplicon junctions in ecDNAs [43,53,60,65]. The development of WGS technologies marked a significant breakthrough in the study of ecDNAs, offering high-throughput, high-resolution genomic data, providing insights into the organization and composition of ecDNA elements [18,30,66,67]. Subsequently, the advent of third-generation sequencing (TGS) and single-molecule sequencing technologies, such as Nanopore sequencing, have further revolutionized ecDNA research by providing an unprecedented resolution for the analysis of complex extrachromosomal structures. Interestingly, such approaches could be applied after the enrichment of ecDNAs to increase the NGS resolution and the specificity of the structural variation (SV) detection. 

One of the earliest methods for enriching the DNA samples of ecDNAs was set up by Radloff et al. in 1967 [68], which used cesium chloride (CsCl)/ethidium-bromide gradients to isolate extrachromosomal circular DNAs (eccDNAs) from the linear DNA. This approach was further integrated with exonuclease V (exoV) treatment to enzymatically digest linear DNA and a Tn5 transposition fragmentation and tagging system for the genesis of sequencing libraries, which were specifically enriched with eccDNAs for their profiling (Circulome-seq) [69]. A pretreatment step consisting of a Plasmid-Safe DNase digestion of linear chromosomes was used in the Circle-Seq approach [70]. EcDNAs were then enriched via rolling circle amplification mediated by Phi29 DNA polymerase. Then, CIDER-Seq (circular DNA enrichment sequencing) aimed to isolate eccDNA for PacBio long-read sequencing without undergoing a PCR or restriction digestion pretreatment. The obtained full-length sequences were subsequently processed using DeConcat, a new read-processing algorithm that does not need any sequence assembly steps [71]. A further refinement of this procedure developed a new three-step eccDNA purification (3SEP) procedure, leading to eccDNA preparations with a high purity and reproducibility level for sequencing purposes, combining rolling-circle amplification with Nanopore sequencing [72].

Hung et al. proposed a new method based on a modified CRISPR-Cas9-Assisted Targeting of Chromosome segments (CATCH) technique, combining the CRISPR-Cas9 treatment and pulsed-field gel electrophoresis (PFGE) of agarose-entrapped genomic DNA to isolate and analyze large ecDNAs, along with the corresponding chromosomal region. This method allowed an ecDNA analysis at single nucleotide resolution, identifying, for example, exclusive oncogene mutations on ecDNAs, such as the EGFRvIII mutation in the glioblastoma GBM39 cell line. The high-resolution potential of this method could also be exploited to study the mechanism of FA genesis, since chromosomal deletions of the corresponding amplified sequences, or “scar”, can be detected, supporting an excision model for ecDNA formation. Moreover, CRISPR-CATCH, followed by nanopore sequencing, could be used for ecDNA cytosine methylation profiling [73].

Bioinformatic tools have been used to analyze the next-generation sequencing data from these experimental methods. Deshpande and colleagues developed AmpliconArchitect (AA), a tool designed to reconstruct the fine structure of focally amplified regions using WGS data. They extensively validated AA on multiple simulated and real datasets, spanning a wide range of coverage and copy numbers. The analysis of 68 cancer samples caused by viral infections revealed FA occurrence in a broad range of cancer types and suggested their involvement in the genesis of complex rearrangements [74]. Additionally, in 2019, Circle-Map was developed by Prada-Luengo et al. as a sensitive method for detecting circular DNA from circle-enriched next-generation sequencing data at single-nucleotide resolution. Circle-Map addresses the limitations of short-read mappers by guiding the realignment of partially aligned reads using information from discordantly mapped reads. This approach significantly enhanced the sensitivity for detecting circular DNA in simulated and real data while maintaining a high precision [75].

Building on AA, in 2020, Luebeck et al. developed AmpliconReconstructor (AR) to integrate optical mapping and next-generation sequencing for resolving focal copy number amplifications at single-nucleotide resolution. AR can detect ecDNAs in WGS data generated using methods such as conventional WGS, Nanopore, and PacBio technologies, but it is most effective when these WGS data are produced from methods that capture long DNA fragments, as AR employs a graph-based approach to identify ecDNAs, which works better with longer DNA fragments [76]. Lastly, in 2022, Mann et al. introduced the ECCsplorer pipeline, designed to detect extrachromosomal eccDNAs in any organism or tissue using Illumina-sequencing-based next-generation sequencing techniques. The pipeline combines read mapping and a reference-free comparison of the read clusters, showcasing its sensitivity and specificity by successfully detecting mitochondrial mini-circles and retrotransposon activation in various organisms. ECCsplorer is valuable for diverse downstream investigations, including cancer-related eccDNAs, organelle genomics, and active transposable elements [77]. 

Importantly, all the resulting bioinformatic data on EccDNAs are collected on specific platforms such as eccDNAdb [78], CircleBase [79], and TeCD (The eccDNA Collection Database) [80], helping researchers to gain information on isolated eccDNA, mainly in human cancers, and investigate their functions. 

The advantages and limitations of each method are summarized in Table 1.

### 3.2. HSR/ecDNA Chromatin Status Assessment

To further elucidate the role of the chromatin structure and histone modifications in ecDNA-driven gene overexpression, researchers have set up various methods over the years (Table 1). ChIP-seq, 4C-seq, and ATAC-seq were introduced in 2007, 2011, and 2013, respectively [93,94,95]. These techniques have enabled the characterization of histone marks, chromatin accessibility, and other epigenetic features associated with ecDNA [30]. For instance, H3K4me1/H3K27ac ChIP-seq has identified multiple active histone marks and a few repressive histone marks on the ecDNA in GBM cells [14]. Furthermore, H3K27ac ChIP-seq has shown an interaction between amplified super-enhancers, downstream mapping the *MYC* gene with the *MYC* promoter in lung adenocarcinomas and endometrial carcinomas [96]. The same approach unveiled interactions between super-enhancers and focally amplified oncogenes in multiple tumors [97]. ATAC-seq shows open chromatin regions due to their sensitivity to the hyperactive transposase Tn5 [86]. This technique has been used to identify eccDNAs in ovarian cancer, prostate cancer, and GBM cell lines, apart from GBM and glioma samples from The Cancer Genome Atlas (TCGA) [87]. ATAC-seq has also been used to study mutations in open chromatin regions and their association with genomic amplification [98]. ATAC-seq is implemented using a modified Tn5 using DNA adapters conjugated to fluorophores. This technique, called an “assay of transposase-accessible chromatin with visualization” (ATAC-see), introduced in 2016, allows for both the sequencing and in situ visualization of the accessible chromatin through the fluorescent signal. This approach could be exploited to study the chromatin accessibility in specific regions of the genome, including those associated with FAs, and to study the mechanisms that regulate the formation and stability of FAs [88].

Furthermore, 4C-seq, a method detecting the chromatin interaction between one single locus and the rest of the genome, has disclosed the highly accessible chromatin of ecDNAs and their long-range interactions with active chromatin, promoting oncogene overexpression in prostate cancer, breast cancer, GBM, medulloblastomas, neuroblastomas (NB), and Wilms tumors [97,99,100]. Proximity ligation-assisted ChIP-seq (PLAC-seq), introduced at the end of 2016, combines ChIP with proximity ligation and high-throughput sequencing. PLAC-seq allows for the study of protein–DNA interactions (e.g., transcription factor binding), histone modification, and DNA methylation. It has also been used to detect and analyze the presence of FAs in tumor biopsies [101]. Along with wet techniques, some bioinformatic tools have been developed to investigate the chromatin structure and its interactions. For example, ChIA-PET, one of the first bioinformatics tools developed in 2010, was designed to automatically analyze the chromatin interactions with paired-end tag sequencing, enabling the investigation of the 3D organization of the genome [102]. In Zhu et al. (2021), the ChIA-PET assay and ChIA-PET Utilities, an implemented version of the ChIA-PET Tool, were used to examine the ecDNA-mediated chromatin contacts in GBM neurospheres and prostate cancer cell cultures. In the same work, ChIA-Drop assays were also performed [92]. This method, introduced in 2019, and the bioinformatics analysis tool, ChIA-DropBox, allows for the study of the interactions between different genomic regions, including those associated with ecDNAs [103]. Integrating ChIA-PET and ChIA-Drop data, researchers have proposed that ecDNAs may function as mobile transcriptional enhancers by recruiting RNA Polymerase enzymes and transcriptional factors that could act in trans on the gene mapping on cell chromosomes, which may also promote tumor progression through this peculiar transcription control mechanism [92].

### 3.3. Analysis of the Transcriptional Landscape Associated with FAs

RNA sequencing (RNA-seq) has been extensively utilized to explore the transcriptomic consequences of genomic amplifications, including those associated with FAs. Various studies have demonstrated that FAs lead to the overexpression of genes promoting tumor progression, increasing genetic heterogeneity and impacting the transcriptional programs in cancers [92,104,105]. By integrating WGS and RNA-seq data from several cancer cell lines and TCGA patients, Wu et al. demonstrated that ecDNA-embedded oncogenes were the most expressed genes when amplified [14]. Intriguingly, Hung et al. revealed that the overexpression of ecDNA-harbored oncogenes was not only promoted by an increase in their copy number, but by trans-acting enhancers mapping on different ecDNA molecules. This event is a result of the clustering of ecDNAs in the cell nuclei, tethered by DNA binding proteins, such as the BET proteins in the colon cancer (CRC) cell line COLO320-DM [19].

FAs with complex structures are also linked to the genesis of fusion genes, joining genes from the same or different chromosomes [21,22,106]. To detect the chimeric RNAs transcribed from ecDNA molecules with a single-cell resolution, a recently introduced method is single-cell extrachromosomal circular DNA and transcriptome sequencing (seEC&T-seq) developed by Gonzales et al. [55]. This approach consists of simultaneously sequencing ecDNAs and mRNA from a single cell; hence, it can identify both the ecDNA SVs joining two genes and their resulting fusion RNAs [55]. Moreover, this method allows for studying ecDNA-embedded oncogene expression and can document the high heterogeneity often accompanying ecDNAs among different cells [55]. 

Various fusion genes are described as being amplified in human cancer, such as *PAX7::FOX01* and *PAX3::FOX01* in alveolar rhabdomyosarcoma [107], *BCR::ABL1* in chronic myeloid leukemia [45], *COL1A1::PDGFB* in dermatofibrosarcoma protuberans [108], *NUP214::ABL1* in T-cell ALL [109], and *PVT1* chimeras in solid tumors and hematological malignancies [110]

There is much debate in the literature about the role of chimeric genes harbored by ecDNAs and HSRs: according to some authors, these chimeras have a passenger role in cancer genesis and progression [111,112]. However, we have recently demonstrated an oncogenic role in vitro for the *PVT1::AKT3* chimera, transcribed from a fusion gene mapping on an HSR in the *MYC*-amplified SCLC cell line GLC1HSR [22]. Indeed, its silencing decreased cell proliferation and increased apoptosis [113]. 

Amplified gene transcripts can also be involved in the genesis of fusion RNAs without genomic support (e.g., joining two RNAs from amplified genes not interrupted by SVs or involving partners transcribed from not amplified genes), as reported by integrating the WGS and RNA-seq data from 8q24-amplified AML patients [21]. These chimeras could be derived from post-transcriptional events, such as trans-splicing or cis-splicing between adjacent genes (cis-SAGe), disclosing a high plasticity of the transcripts from amplified genes [114,115].

FAs can also lead to the overexpression of circular RNAs (circRNAs). 8q24 amplification, harboring *PVT1* exon 2, could lead to hsa_circ_0001821 (circPVT1) overexpression, as documented in AML patients with *PVT1*-harboring DMs/HSRs in comparison to samples with a normal karyotype or an amplification of different genes [21]. circPVT1 has been reported as having an oncogenic role in several cancer types [116]. Hence, its overexpression, linked to its genomic locus amplification, deserves further investigation. Moreover, this result shows that the transcriptome of cancer cells with FAs can also become more complicated due to the overexpression of oncogenic circRNAs from amplified genes. By shedding light on these complexities, RNA-seq illustrates its indispensable role in the characterization of the FA-driven cancer transcriptional landscape, thus deepening our understanding of the role of FAs in tumorigenesis and guiding future therapeutic strategies [14,19,92,104,105].

## 4. Methods to Detect Genomic Amplification in Liquid Biopsies

Advances in the detection of cell-free DNA (cfDNA), circulating tumor cells (CTCs), and Extracellular Vesicles (EVs) in the blood, as well as in other body fluids (urine, saliva, and sperm), have improved the non-invasive testing potential for cancer diagnoses and follow-ups.

Applying such approaches for eccDNA detection in liquid biopsies would represent an important new instrument, not only for the rapid detection of cancers harboring genomic amplification, but also for patient response monitoring after a specific therapy to promptly identify the development of possible resistance mechanisms. Moreover, as genomic amplification contributes significantly to tumor heterogeneity, which is the main cause of a lack of response to cancer therapies [18], the setup of circulating eccDNA detection methods would offer novel biomarkers that could be used in clinical practice more effectively than traditional molecules. Much effort should be spent to ensure the protocols dedicated to detecting eccDNA in liquid biopsies are more applicable at a large-scale level, particularly to improve the sensitivity and specificity of their identification.

Unfortunately, the literature on the early detection of eccDNA in liquid biopsies is limited to a few current reports.

In particular, the release of eccDNAs has been documented in the blood circulation and urine of humans [117,118]. The presence of eccDNA of maternal and fetal origin has also been documented in the plasma of pregnant women [119]. In cancer patients, eccDNA is detected at high levels in their serum and plasma before surgery and at lower levels 15 days after a tumor resection surgery [120]. Therefore, their detection and investigation via molecular approaches aimed at identifying the amplified oncogenes on ecDNAs could be a promising approach for diagnosing and monitoring cancer in clinical practice. 

A method for detecting and characterizing eccDNAs in normal and tumor plasma samples employed the treatment of cfDNA with ATP-dependent DNase digestion to remove the linear DNA and subsequent Multiple Displacement Amplification (MDA), in order to amplify the undigested circular DNAs. The amplified DNAs were then used to prepare sequencing libraries and a custom program called “split-align” was utilized to identify the sequences with split reads, which are indicative of eccDNAs [117,120]. 

Interestingly, recent evidence has indicated that a ChIP-seq analysis of the H3K36me3 histone modification from plasma cell-free nucleosomes (cfChIP-seq) was successfully performed on samples from metastatic colorectal carcinoma (mCRC), non-small-cell lung cancer (NSCLC), and SCLC patients, identifying amplified oncogenes, such as *HER2* in mCRC patients, as showing the highest H3K36me3 enrichment [56,121]. These results suggest that ecDNAs can non-invasively be detected in patient plasma, opening up new scenarios for the diagnostic approaches to mCRC, NSCLC, SCLC, and other cancer types.

## 5. Clinical Significance of Genomic Amplification Detected in Tumor Samples

The clinical significance of FAs in tumor samples has emerged as a critical area of investigation in cancer research, with an impact on cancer diagnosis, prognosis, and treatment (Figure 4) [18]. 

Identifying FAs and their contribution to increased gene copy numbers has paved the way for advancements in cancer diagnostics, mainly for early tumor diagnosis and patient stratification for more effective treatment. Indeed, FAs can identify specific cancer subtypes. For example, *HER2*-amplified breast cancers, characterized by aggressive tumor growth and a high tumor recurrence, specifically respond to trastuzumab and pertuzumab monoclonal antibodies [122]. Both therapies have been shown to impact the survival rates and disease progression in patients, highlighting the importance of accurately identifying such FAs upon diagnosis. In addition, as already described in the previous paragraph, non-invasive approaches focused on identifying the circulating ecDNAs in liquid biopsies could offer new diagnostic tools for detecting early-stage tumors. However, these approaches are still far from being used in clinical practice.

FAs have been shown to influence the prognoses of various cancer types. In NB, FA-driven *MYCN* amplification is one of the genomic aberrations that has been reported in high-risk patients, associated with a poor overall survival [123]. Moreover, among *MYCN*-amplified patients, those also showing rearrangements involving amplified chromosomal regions present a worse overall survival than patients lacking such additional alterations [30]. In GBM, FAs carrying *EGFR* and its variant *EGFRvIII* play a crucial role in patient prognoses and targeted therapy resistance. Nathanson et al. [9] demonstrated that *EGFR* amplification mediated by FAs is associated with a worse overall survival and higher tumor recurrence. Their study revealed that the dynamic regulation of extrachromosomal mutant *EGFR* DNA contributes to the development of resistance to targeted therapy, emphasizing the need for new strategies to overcome these challenges in GBM treatment. Despite growing evidence supporting the predictive value of FAs, several challenges remain. The heterogeneity of FAs and their variable presence across cancer types and subtypes necessitate further investigation to establish their clinical utility as reliable prognostic markers [124]. Additionally, developing standardized methods for detecting and quantifying FAs in clinical samples will be crucial for their implementation in routine clinical practice [14].

As our knowledge of FAs constantly evolves, exploring their potential applications as therapeutic targets becomes increasingly important. For example, recent studies have demonstrated that eliminating FAs in NB can enhance drug sensitivity [125]. One strategy for targeting FAs involves disrupting their maintenance and replication in cancer cells. Several studies have demonstrated the potential of various treatments to eliminate or reduce the extrachromosomal DNA and amplified oncogenes in different types of cancer cells, such as hydroxyurea [126,127], inhibitors of poly (ADP-ribose) polymerase (PARP) and dimethyl sulfoxide [128], and ionizing radiation [129]. A study by Von Hoff et al. showed that treating human tumor cells with low concentrations of hydroxyurea accelerated the loss of extrachromosomally amplified oncogenes, such as *MYC*, which was correlated with a dramatic reduction in tumorigenicity [127]. Moreover, in CRC, using DNA-PKs or PARP inhibitors affecting NHEJ decreases the chromothripsis-mediated genesis of DMs [104]. Similarly, Meng et al. found that inhibiting NHEJ can prevent the formation of DMs in methotrexate-resistant CRC [130]. Another approach to exploiting FAs in cancer therapy is to target the oncogenic signaling pathways activated by amplified oncogenes. For example, in CRC, *MET* amplifications have been associated with a poor prognosis and resistance to anti-EGFR therapies, including the anti-EGFR cetuximab monoclonal antibody [131,132]. Bardelli and colleagues found that *MET* inhibitors, such as crizotinib and cabozantinib, restored the sensitivity to cetuximab in *MET*-amplified patient-derived CRC xenografts [131]. Another example regards *MYC*-amplified tumors: in this case, using *MYC* small-molecule inhibitors, such as BET inhibitors, could be a more effective treatment strategy than conventional chemotherapy [133]. 

FAs can also inform personalized treatment decisions by predicting therapy response and resistance. Tumors harboring FAs are often characterized by an increased genetic heterogeneity, which can lead to therapy resistance [134]. For example, in NSCLC, FA-driven *EGFR* amplification has been associated with a resistance to *EGFR* tyrosine kinase inhibitors, such as erlotinib [135]. In these cases, alternative targeted therapies, such as osimertinib, which has shown efficacy in patients with T790M-mutated *EGFR*, may be more effective [136]. In *MET*-amplified NSCLC, *MET* amplification represents a resistance mechanism to *EGFR* tyrosine kinase inhibitors. *MET*-targeted therapies, such as gefitinib, could represent the best therapeutic choice for these patients [137]. FA events also play a crucial role in shaping the immunotherapy decision making in various cancers. Zhang et al. [138] found that *MET* amplification in lung cancer patients attenuated the tumor response to immunotherapy by inhibiting the STING levels and reducing the antitumor T-cell infiltration, suggesting that combining *MET* inhibitors with an immune checkpoint blockade could help overcome resistance. Deng et al. [139] discovered that a RAD21 amplification in ovarian cancer led to the epigenetic suppression of interferon signaling, promoting immune evasion and highlighting *RAD21* as a potential target and biomarker for precision immunotherapy. Xu et al. [140] investigated the genomic and transcriptional heterogeneity of multifocal hepatocellular carcinoma, revealing that it could influence the clinical responsiveness of patients to targeted drugs and immunotherapies, underlining the importance of personalized treatment strategies. Finally, in Massó-Vallés et al. [141], the potential targeting of *MYC*, *MYCN*, and *MYCL* in lung cancer was discussed, emphasizing that a combination of *MYC* inhibitors with immunotherapies might represent a promising strategy for improving treatment outcomes, particularly given the role of *MYC* in immune suppression.

## 6. Conclusions

Despite the potential clinical implications of FAs in cancer, numerous challenges persist, including developing selective and effective FA-targeting therapeutic agents. For this reason, a deeper understanding of the molecular mechanisms responsible for FA genesis, maintenance, and replication is required. For instance, investigating chromothripsis and the role of replication stress in FA formation can unveil unique DNA repair vulnerabilities and druggable targets in cancer cells. Advanced technologies, such as single-cell and long-read sequencing and RNA-seq-based techniques, will enhance the detection and characterization of FAs, allowing for a more comprehensive understanding of their role in cancer biology and the development of efficient therapeutic strategies. However, the potential off-target effects and toxicity of FA-targeting therapies should also be rigorously assessed to ensure patient safety.

Moreover, improving the methods and tools for detecting FAs in liquid biopsies could offer significant advantages in cancer diagnoses and patient follow-ups, such as non-invasive sample collection, the real-time monitoring of treatment responses, and the ability to detect minimal residual disease.

For all the reasons mentioned above, ongoing investigations into FA are indispensable for disclosing the potential of this powerful tool in the fight against cancer.

## Figures and Tables

**Figure 1 genes-14-01304-f001:**
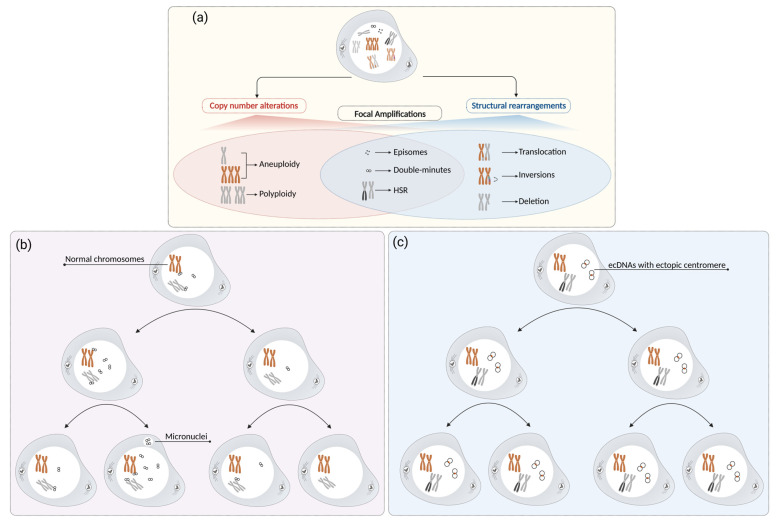
FAs and their inheritance through cell generations: (**a**) overview of different FA types; (**b**) segregation of ecDNAs (DMs) during cell division. The lack of centromeres in ecDNAs results in unequal segregation into daughter cells. Throughout mitosis, ecDNAs attach randomly to the ends of chromosomes (chromosome tethering) and distribute asymmetrically between daughter cells. As anaphase progresses, those ecDNAs not attached to chromosomal extremities accumulate, fail to be incorporated into emerging daughter nuclei, and may persist within the micronucleus in a few cells; and (**c**) segregation of normal chromosomes, HSRs, and ecDNAs with ectopic centromeres during cell division. Ectopic centromeres maintain a segregation pattern similar to that of conventional chromosomes, thereby determining their mitotic stability.

**Figure 2 genes-14-01304-f002:**
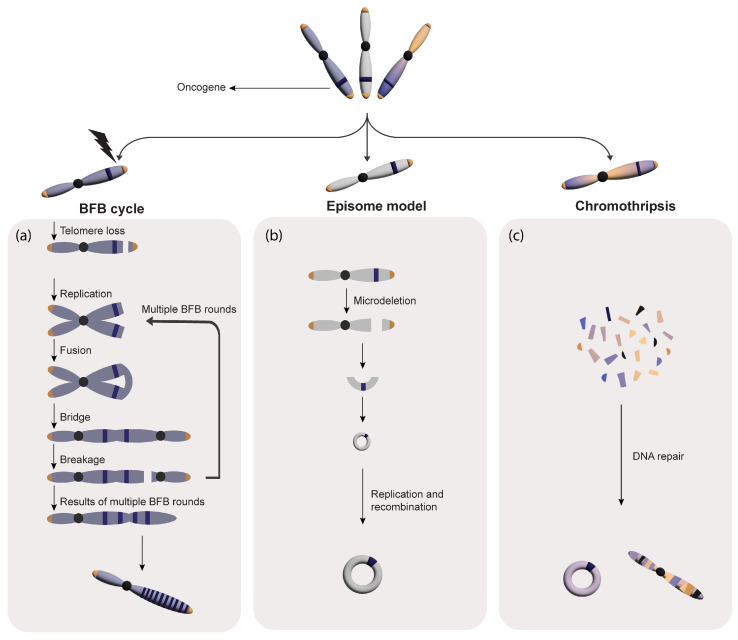
Potential models of FAs biogenesis: (**a**) BFB Cycle: a double-strand break occurring downstream of a known oncogene may lead to the fusion of broken chromatids, forming a dicentric anaphase bridge. This telomere-devoid bridge undergoes breakage at cytokinesis, leading to the genesis of head-to-head amplicon copy number increase through the repetition of breakage-fusion-bridge cycles; (**b**) Episome Model: Replication fork stalling or loop formations may induce the excision of genomic regions, harboring oncogenes, that undergo circularization through ligation of their extremities. The newly originated episomes can evolve into ecDNAs through replication and recombination; and (**c**) Chromothripsis Model: Chromosome shattering can yield multiple acentric DNA segments. The DNA repair machinery reassembles some fragments, originating ecDNAs and new chromosomes featuring complex structural rearrangements. Occasionally, selective pressure and linear DNA damage may prompt ecDNA molecules to reintegrate into the linear genome, originating HSRs.

**Figure 3 genes-14-01304-f003:**
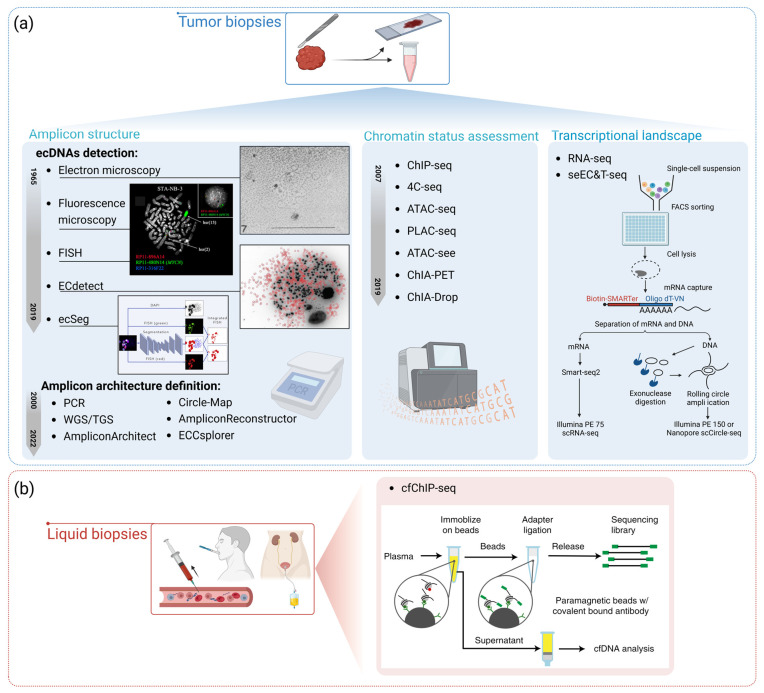
Timeline and overview of methodologies for investigating FAs in tumor and liquid biopsies: (**a**) A chronological overview of wet-lab techniques and bioinformatics tools developed for analyzing the amplicon unit structure, chromatin architecture, and transcriptional landscape associated with FAs, underlining the significance of an integrated multi-technique approach in this domain [14,18,53,54,55]; and (**b**) cfChIP-seq technique as a valuable technique for detecting FAs in liquid biopsies [56].

**Figure 4 genes-14-01304-f004:**
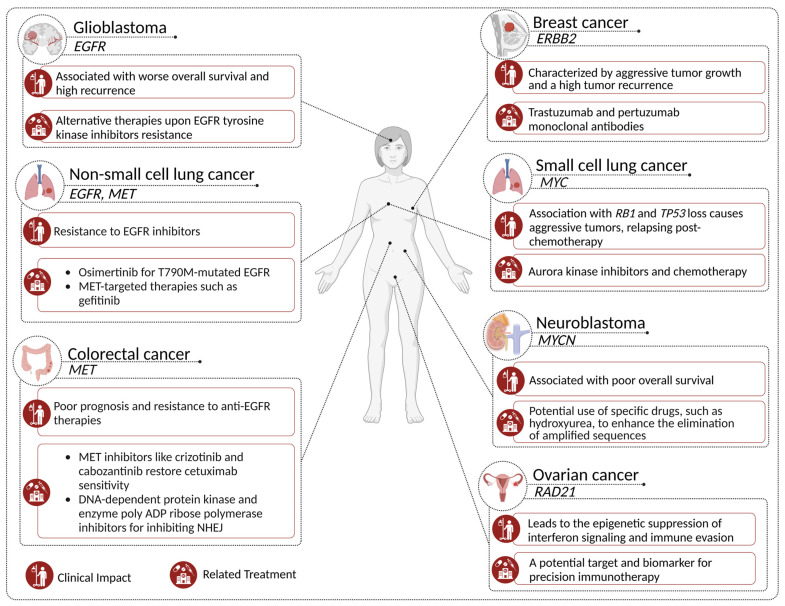
Representative impact of specific gene amplifications on clinical outcomes and corresponding treatment strategies in different types of solid cancers.

**Table 1 genes-14-01304-t001:** Methods for amplicon analysis and their advantages/disadvantages.

Method for Amplicon Analysis	Gained Information	Strengths	Limitations	References
Southern Blotting	Validation of FA breakpoints	Direct analysis of cell DNA	Preliminary information on the amplified regions is required to design appropriate probes	[64]
FISH	FA visualization within cell nuclei by the use of specific fluorescent probes	Identification of FA cytogenetic manifestation (ecDNAs vs. HSRs); definition of HSR insertion sites; detection of cell heterogeneity and number of FAs per cell	Resolution linked to the probe size (ranging from the whole chromosome to 40–50 Kb)	[43]
Long-range PCR approaches	Characterization of amplicon SVs and HSR insertion sites	Single nucleotide resolution analyses; informative to the mechanisms involved in FA genesis (e.g., NHEJ)	Need of preliminary information on the amplified regions to design appropriate primer pairs	[43,53,60,65]
WGS-based technologies	Single nucleotide level characterization of FAs	High-resolution, high-throughput analyses; reconstruction of the whole amplicon structure	Issues in discriminating between ecDNAs and HSRs; problems in validating rearranged ecDNA sequences due to the read length	[18,30,66,67]
Circulome-seq	Enrichment and sequencing of ecDNAs	Good yield also starting from a small amount of ecDNAs; detection of ecDNAs up to several hundred kb	Impossibility to obtain single-molecule sequence data	[69]
Circle-Seq	Enrichment and sequencing of ecDNAs	Suitable for analysis of broad-range size ecDNAs	Potential incomplete removal of linear DNA by exonuclease digestion	[70]
CIDER-Seq	Purification and sequencing of ecDNAs	No need for PCR assays, cloning, and computational sequence assembly	High accuracy only for ecDNAs smaller than 10 kb; need for long-read sequences	[71]
3SEP	Purification of ecDNA molecules	Accurate removal of linear DNA and mitochondrial genome; time-saving technique	Limited to small ecDNA molecules	[72]
CRISPR-CATCH	Sample enrichment with ecDNAs and the corresponding genomic locus	Large size ecDNA enrichment; possibility to isolate and analyze the corresponding chromosomal locus of the amplified DNA; ecDNA separation based on the size	Need of preliminary information on the amplified regions to design appropriate sgRNAs	[73]
AmpliconArchitect	WGS-data-based reconstruction of the internal structure of FAs with single-nucleotide resolution	Characterization of complex and heterogeneous amplicons	Issues in FA reconstruction in presence of duplicated segments within the amplicons	[74]
Circle-Map	Fine detection of ecDNAs and breakpoint junctions of the circular DNA structure	Highly sensitive and precise tool	Need of circle-enriched data to produce the most accurate results; impossibility to detect SVs internal to the FA structure	[75]
Amplicon Reconstructor	Single-nucleotide resolution reconstruction of the fine-scale and large-scale FA architecture, based on the optical mapping of long DNA fragments and NGS data	Reconstruction of amplicons with complex architectures, mainly due to the use of long-range sequencing data	Limitations of the optical map assembly process; issues in reconstructing the FA structure when nested duplication of amplicon segments occur	[76]
ChIP-seq	Information on FA chromatin accessibility	Identification of actively transcribed chromatin at a single-nucleotide resolution; possibility to perform single cell analyses	Introduction of a potential bias by PCR amplification; potential for epitope masking due to the formaldehyde crosslinking process	[81]
4C-seq	Unbiased detection of long-range chromatin interactions involving amplified regions	High-throughput screening of physical interactions between chromosomes without pre-determination of the interacting partners	Missing local interactions (<50 kb) from the region of interest; not efficient amplification of large circles	[82,83]
ATAC-seq	Detection and mapping of FA chromatin accessibility regions	Gather chromatin regions of increased accessibility;map regions of transcription-factor binding and nucleosome position; detect ecDNAs at the pre-amplification stage, useful for predicting resistance to therapy	Potential contamination of data with mitochondrial DNA; potential sequence or structural biases of the Tn5 enzyme	[84,85,86,87]
ATAC-see	Unveiling of the specific spatial organization of the accessible FA chromatin, labelling open chromatin accessible loci	Revelation of the accessible chromatin regions in their native context; analysis of fixed samples, providing both spatial and epigenomic information	Potential contamination of data with mitochondrial DNA; potential sequence or structural biases of the Tn5 enzyme	[88]
ChIA-PET	Study of the ecDNA chromatin interactions within its structure and with chromosomal regions and ecDNA sequences	Suitable for detecting a large number of both long-range and short-range chromatin interactions globally, independently from their size and sequence content; removing background generated during traditional ChIP assays; best resolution and coverage balance to map long-range interactions	A large amount of cell material required, generally (at least 10^8^); limited sensitivity	[82,89,90,91]
ChIA-Drop	Characterization of the ecDNA-associated chromatin interactions	Decipher ecDNA-chromosome complexity at a single-molecule resolution; direct analysis of chromatin samples without the need to purify them		[92]

## Data Availability

Not Applicable.

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
