# Peer review of "Advancements in Focal Amplification Detection in Tumor/Liquid Biopsies and Emerging Clinical Applications"

_genes, 2023, doi:10.3390/genes14061304_

Round 1

Reviewer 1 Report

Review Report for the Manuscript “Advancements in focal amplification detection in tumor/liquid biopsies and emerging clinical applications”

Rating the Manuscript

English Level: Is the English language appropriate and understandable?

Yes, English language in the manuscript is appropriate and understandable. 

General comments

This manuscript is well written. Figures are good and they are very informative. Figure captions are also very descriptive. I only have few comments on this, and they are given below.

Overall Recommendation: 

Accept after Minor Revisions

Given below are the comments for each section of the manuscript.

Abstract

Line 14:

“Numerous wet-lab and bioinformatics approaches have been set up to detect FAs, unravel the internal structure of amplicons, assess their chromatin compaction status, and investigate the transcriptional landscape associated with their occurrence in cancer cells.”

Mention few wet-lab approaches that have been used for detecting FAs.

Line 20: 

“Despite the potential therapeutic implications of FAs, challenges remain, including developing selective and effective FA-targeting agents and understanding the molecular mechanisms underlying FA maintenance and replication.”

What are the therapeutic implications of FAs? Name a few here.

1.Introduction

Line 38:

“They are present in at least 30 different cancer types, often associated with a worse patient prognosis.”

Mention few cancer types in which FAs occur.

2. Cytogenetic manifestation of genomic amplifications and mechanisms of their genesis

2.1. Types of genomic amplification

Line 116: 

“Despite the established acentric nature of DMs, we documented the emergence of ectopic centromeres (neocentromeres) at them, increasing their size towards small ring chromosomes in a portion of tumor cells in the bone marrow of Acute Myeloid Leukemia (AML) patients with MYC-DM amplifications”.

Is it only present in AML patients or is it common to all the cancer patients?

3. Methods and bioinformatic tools to detect genomic amplification in tumor samples

3.1. Definition of amplicon structure at the DNA level

3.1.1. Detection of DMs/HSR

I think it would be better if the authors could briefly discuss the advantages and disadvantages of these detection methods.

Line 231:

“Early studies on FAs relied on electron microscopy (EM), including scanning electron microscopy (SEM) and transmission electron microscopy (TEM), providing the first visual evidence of FAs in cancer cells and particularly useful in visualizing the circular nature of DMs”.

It would be informative if the authors could include some images that were acquired by these detection methods. They could include these in Figure 3.

3.1.2. Amplicon architecture definition and 3.2. HSR/ecDNA chromatin status assessment

I think it's better and easy to follow if the authors include a table describing advantages and disadvantages of each of these approaches and discuss what information could be obtained by each method.

4. Methods to detect genomic amplification in liquid biopsies

Line 434:

“In particular, the release of eccDNAs has been documented in the blood circulation and urine of humans. The presence of eccDNA of maternal and fetal origin was also documented in the plasma of pregnant women.”

Is there any published data that directly compares the results from tumor biopsies and liquid biopsies. 

References

Some of the references are more than 10 years old. If these don’t contain important information, authors need to replace these references with newer ones.

References:

1,2,3,4,5,6,7,8,9,19,21,22,23,30,31,32,33,37,41,42,43,44,46,47,48,49,50,51,52,53,56,57,58,59,60,61,63,64,67,81,82,93,99,101,103,104,106,109,110,121,122,123,124,125,126,130,131,133,135,

Reviewer 2 Report

Overall, the manuscript discusses the focal amplification (FA) types, probable causes, and other aspects in tumor samples and liquid biopsies as novel methods for cancer detection. The content is well-organized, and the figures are of good quality. However, here are some suggestions for improvement:

Summarize the different types of FA in figures: The authors could provide a clear summary of the different types of FA in the figures. This would help readers better understand FA and allow for comparisons with other mutations or copy number alterations. By visually presenting the information, the authors can enhance comprehension and facilitate comparisons.

Summarize the roles of FA in different cancers: To provide a comprehensive understanding of FA, it would be beneficial to summarize its roles in different types of cancers. This could be done using a table or figures that illustrate the impact of FA on diagnosis, prognosis, therapy, and other relevant aspects. Such a summary would enable readers to grasp the diverse implications of FA across various cancer types.

Expand on Chapter 4: The authors should consider providing more details about Chapter 4, specifically focusing on the advantages of FA detection in liquid biopsies compared to current biomarkers. Liquid biopsies offer unique opportunities for non-invasive cancer detection and monitoring, and highlighting the specific advantages of FA detection in this context would be valuable. Discussing the increased sensitivity, specificity, or early detection capabilities of FA in liquid biopsies compared to traditional biomarkers could strengthen the manuscript.

By addressing these suggestions, the manuscript will be more informative and provide a clearer understanding of focal amplification, its relevance in different cancer types, and the advantages of FA detection in liquid biopsies.

None
